# Synthesizing Polyurethane Using Isosorbide in Primary Alcohol Form, and Its Biocompatibility Properties

**DOI:** 10.3390/polym15020418

**Published:** 2023-01-12

**Authors:** Suk-Min Hong, Hyuck-Jin Kwon, Chil-Won Lee

**Affiliations:** Department of Chemistry, College of Science and Technology, Dankook University, Cheonan 31116, Republic of Korea

**Keywords:** polyurethane, biocompatibility, polycarbonate diol, bis(2-hydroxyethyl)isosorbide, one-shot polymerization

## Abstract

Isosorbide is a bio-based renewable resource that has been utilized as a stiffness component in the synthesis of novel polymers. Modified isosorbide-based bis(2-hydroxyethyl)isosorbide (BHIS) has favorable structural features, such as fused bicyclic rings and a primary hydroxyl function with improved reactivity to polymerization when compared to isosorbide itself. Polyurethane series (PBH PU series) using polycarbonate diol (PCD) and bis(2-hydroxyethyl)isosorbide (BHIS) were polymerized through a simple, one-shot polymerization without a catalyst using various ratios of BHIS, PCD, and hexamethylene diisocyanate (HDI). The synthesized BHIS and PUs were characterized using proton nuclear magnetic resonance (^1^H-NMR), Fourier transform infrared (FT-IR), differential scanning calorimetry (DSC), and mechanical testing. To determine the feasibility of using these PUs as biomedical materials, we investigated the effects of their BHIS content on PBH PU series physical and mechanical properties. The PBH PU series has excellent elasticity, with a breaking strain ranging from 686.55 to 984.69% at a 33.26 to 63.87 MPa tensile stress. The material showed superb biocompatibility with its high adhesion and proliferation in the bone marrow cells. Given their outstanding mechanical properties and biocompatibility, the polymerized bio-based PUs can contribute toward various applications in the medical field.

## 1. Introduction

Over the recent years, research on bio-based polyurethane has been concentrated. These studies focus on the energy consumption decrease, greenhouse gas emissions, and CO_2_ decrease in bio-based monomers and polymer production [1].

Polyurethane (PU) is flexible, high in strength, and can be used in a wide range of applications, including foams, coatings, sealants, adhesives, and elastomers [1,2,3,4,5,6]. PU consist of polyol and isocyanate, and its softness, hardness, and flexibility can be adjusted. Polyol is a soft segment region, and isocyanate is a hard segment region, and so PU can be applied to various fields by adjusting the ratio according to the desired use [7]. PU is also used for biomedical applications owing to its wide range of physical properties, easy processability, and excellent biocompatibility [8,9,10]. Over the past several decades, the applications of PU have expanded with the introduction of biomolecules (polysaccharides, lipids, and proteins) to improve its renewable properties and biocompatibility [11,12].

Due to their abundance, high functional diversity, and biocompatibility, carbohydrate derivatives are increasingly used in polymer synthesis [13,14]. The 1,4:3,6-dianhydro-d-sorbitol (isosorbide, IS) is known as a renewable and biocompatible material [15]. It can be obtained through the simple dehydration of sorbitol [16]. IS is composed of two cis-fused tetrahydrofuran rings. The hydroxyl groups are situated at carbons 2 and 5, which are designated as endo or exo, respectively. The structure is also composed of secondary di-alcohols [17,18]. It is surmised that polymers with a high glass-transition temperature or special optical properties can be synthesized [15]. In addition, due to the harmless properties of the molecule, it can be used as an alternative to bisphenol A and can be applied in packaging and medical devices [19,20,21,22].

Polycarbonate diol (PCD) can be used as a raw material for PU. PCD-based PUs have sufficient mechanical strength, abrasion resistance, and balanced physical properties [23,24]. PCD-based PUs are preferred, especially for applications requiring high mechanical strength [25]. Although hexamethylene diisocyanate (HDI) is less reactive than other diisocyanates, it is one of the most widely used aliphatic diisocyanates because of its resistance to turning yellow, excellent heat resistance, and its demand in various industries [26,27]. Additionally, aliphatic isocyanates were used as PU polymer components due to their biocompatibility without toxic degradation by-products, unlike aromatic isocyanates [28]. The mechanical properties of aliphatic isocyanate-based PU are not as good as those of aromatic isocyanate-based PU, and to improve this, isosorbide with a rigid structure can be used as a chain extender [29]. However, IS with secondary alcohols have a low reactivity, limiting the reaction participation in PU polymerization [15].

To solve the above problem, this study used ethylene carbonate (EC) and potassium carbonate to synthesize primary di-alcohol-type bis(2-hydroxyethyl)isosorbide (BHIS). This study also used PCD, HDI, and BHIS to synthesize bio-based PU. The obtained tensile strain was 984.69 ± 13.79%, and the tensile stress value of 63.87 ± 1.23 MPa was considered to be encouragingly high. Such bio-based PUs have non-toxic and excellent mechanical properties and can be used in bone grafts, as materials to repair articular cartilage and scaffolds, and in various other medical applications.

## 2. Materials and Methods

### 2.1. Materials

Polycarbonate diol (PCD, M_w_: 2000, 98%) was purchased from Asahi Kasei Chemica (Tokyo, Japan). Hexamethylene diisocyanate (HDI, 98%), 1,4:3,6-dianhydro-D-sorbitol (IS, 98%), ethylene carbonate (EC, 99%), potassium carbonate (99%), and silica gel (70–230 mesh) were purchased from Alfa Aesar (Ward Hill, MA, USA). Ethyl acetate (99%), methyl alcohol (99%), N,N-dimethyl-formamide (DMF, 99.9%), and isopropyl alcohol (IPA, 99%) were purchased from Duksan Chem. Co. Ltd. (Seoul, Republic of Korea). The purchased compounds were used directly without further purification. All glassware was oven-dried prior to use.

### 2.2. Synthesis of Bis(2-Hydroxyethyl)Isosorbide (BHIS)

A mixture of 1,4:3,6-dianhydro-D-sorbitol (IS, 100 g, 684.28 mmol) and ethylene carbonate (EC, 132.57 g, 1436.98 mmol) was degassed with nitrogen for 30 min while stirring. After heating at 70 °C for 1 h, potassium carbonate (1134 g, 8.21 mmol) was added, reacted at 170 °C for 48 h, and then cooled to room temperature for 5 h (Figure 1a). The synthesized BHIS was purified by silica column chromatography using methanol/ethyl acetate (1/9) mobile phase to remove potassium carbonate and attain a brown liquid. After evaporation of methanol and ethyl acetate, the residue was subjected directly to Kugelrohr distillation. The experiment yielded 83% BHIS (boiling point: 170 °C (4.0 × 10^−^^3^ mbar)).

### 2.3. Characterization of BHIS

^1^H-NMR spectra of BHIS were measured using nuclear magnetic resonance (500 MHz, Bruker, Karlsruhe, Germany), with 5% (*w*/*v*) polymer solution in Chloroform-d. Gas chromatography mass spectra (JEOL Ltd., Tokyo, Japan) were obtained using a JEOL JMS-700 spectrometer in the electron ionization (EI) mode. The hydroxyl numbers of the BHIS were determined according to the ASTM E 1899-97 standard test method for hydroxyl groups via a reaction with toluene-4-sulfonyl-isocyanate and potentiometric titration with tetrabutylammonium hydroxide. After obtaining the hydroxyl value, the BHIS molecular weight (BHMW) was calculated using the following formula: BHMW = (56,100 mg/hydroxyl value mg KOH × the number of hydroxyl groups in BHIS).

### 2.4. Synthesis of PUs

PUs were synthesized by one-shot bulk polymerization using various ratios of PCD, HDI, and BHIS (Figure 1b). PCD (59.457 g, 29.728 mmol) and BHIS (6.964 g, 29.728 mmol) (reactant stoichiometry and precursor weight are given as an example for PBH PU-2 in Table 1) were used after vacuum drying for 24 h at 80 °C before use. PCD and HDI were placed in a four-neck round-bottom flask (250 mL) under a nitrogen atmosphere, dissolved by stirring at 80 °C. Then, HDI (10 g, 59.457 mmol) was dropped for 3 min while stirring. One-shot bulk polymerization was carried out at 120 °C for 8 h using a mechanical stirrer to prepare PU. The PU was dissolved in DMF (0.2 L) and re-precipitated in IPA (4 L). After reprecipitation, it was vacuum-dried at room temperature for 2 days. The formulation used is reported in Table 1.

### 2.5. Preparation of PU Films

For the degradation, mechanical properties, and cell tests, the PUs were dissolved in DMF at 10 *wt*% and poured into a polytetrafluoroethylene (PTFE) plate. The PU films were dried in a 60 °C vacuum oven for 6 h. Then, after cooling at room temperature, the measured PU films thickness were 0.2 mm (± 0.02).

### 2.6. Characterization of PU Series

The synthetic structure of PUs were measured using nuclear magnetic resonance and Fourier transform infrared. ^1^H-NMR spectra of PUs were measured using nuclear magnetic resonance (500 MHz, Bruker, Karlsruhe, Germany), with 5% (*w*/*v*) polymer solution in Chloroform-d. FT-IR spectra of PUs were measured using a Fourier transform infrared (Varian 640-IR, Varian, Sydney, Australia), and the spectrum ranged from 4000 to 700 cm^−^^1^. The number average molecular weights (M_n_), weight average molecular weights (M_w_), and polydispersity index (PDI) of the PUs were measured by gel permeation chromatography (ACQUITY, Waters, Milford, CT, USA). The measurement conditions were measured using the standard sample polystyrene (M_w =_ 47,200, 129,000 and 264,000) and using the Tetrahydrofuran solution at a rate of 0.5 mL/min^−^^1^. The thermal analysis was measured using a differential scanning calorimetry (Exstar 7020, SEIKO, Tokyo, Japan). The PU sample was sealed in an Al pan, cooled to −70 °C under a nitrogen atmosphere, and then heated to 300 °C at a rate of 10 °C min^−^^1^ and measured. Contact angle equipment (Phoenix 300, Surface Electro Optics, Suwon, Republic of Korea) was used to measure the wettability of the polymer surface.

### 2.7. Mechanical Properties

Tensile strain and tensile stress were measured using Universal Testing Systems (Model 3344, INSTRON, MA, USA). The PU films were cut into dumbbell shapes of 0.2 mm × 5 mm × 50 mm (thickness × width × length). Measurement conditions were measured at a speed of 10 mm/min^−^^1^ at room temperature. The reported results are the mean values for five replicates per experiment [29].

### 2.8. Degradation Test

The degradation of the PU films (25 mm diameter) was measured after drying. Weight (W_0_) of the dry film was measured and the film was dipped in a cylindrical tube (20 mL) with phosphate buffered saline (PBS, pH = 7.3, 10 mL). The experiments were conducted in a water bath (Daihan Wise Bath, Seoul, Republic of Korea) at a temperature of 37.5 ± 0.5 °C. The measurement periods were 1, 5, 10, 15, 21, and 56 days. The samples were rinsed twice with distilled water, dried in a vacuum oven at 40 °C for 2 days, and weighed (W_t_). The final weight (%) was calculated as W_0_ − W_t_/W_0_ [30].

### 2.9. Cell Culture and Cell Viability

The human bone marrow stromal cells (BMSCs) have been purchased from American Type Culture Collection (ATCC, Rockville, MD, USA) and cultured in Dulbecco’s Modified Eagle’s (Thermo Fisher Scientific, Waltham, MA, USA), supplemented with 10% fetal bovine serum (Thermo Fisher Scientific, MA, USA) and 100 units mL^−^^1^ streptomycin/penicillin at 5% CO_2_ and 37 °C. The 2,5-diphenyl-2H-tetrazolium bromide (MTT) assay method was used to measure the cell viability of the PU films. For a short duration, after putting the PU films into a culture plate (24-well), they were washed using Dulbecco’s phosphate buffered saline (pH 7.4). The bone marrow cells were seeded at a density of 2 × 10^5^ cells per well and cultured for a duration of 5 days in the culture medium. Monitoring of MTT assays was performed on 1, 3, and 5 days, and cells were incubated with 300 μL of MTT (0.5 mg mL^−^^1^) for 4 h at 37 °C. After the incubation, MTT was removed and 200 µL of dimethyl sulfoxide was added to each well, and its concentration was measured at 540 nm using an absorbance microplate reader (SpectraMax M2e, Molecular Devices, Silicon Valley, CA, USA). The obtained values were expressed as a fold change. Four independent experiments using four separate wells were performed for the above analysis.

### 2.10. Immunocytochemistry

Cells were treated with a solution of paraformaldehyde (4%) for 30 min at room temperature and then Triton X-100 (0.2%) was added to the PBS/ Bovine Serum Albumin (BSA) solution (2%) for 5 min. The cells were then washed. The cells were then blocked with normal serum (10%) for 1 h and then were incubated with phalloidin-FITC (1:500, Thermo Fisher Scientific, MA, USA). The cells were then diluted in PBS/BSA solution (2%) for 30 min at room temperature and washed 3 times with a PBS solution (0.1 M). The cells were then contrast-stained with 2-(4-amidinophenyl)-1H-indole-6-carboxamidine (Thermo Fisher Scientific, MA, USA), and the slides were mounted in the mounting media (Dako, CA, USA). The mounted slides were observed using a laser-scanning confocal microscope (LSM700, ZEISS, Baden, Germany).

## 3. Results and Discussion

### 3.1. Synthesis and Characterization of BHIS

The BHIS was synthesized using the IS and EC. Potassium carbonate was used as an alkaline catalyst. EC is ring opening, after hydroxy groups on both sides of the IS occur in parallel with carbonyl carbon attack (1) and alkylene carbon attack (2). The carbonyl carbon attack is reversible, creating carbonate groups within the BHIS. Due to the reversibility of the carbonyl carbon attack and the regeneration of the thermally stable EC, carbonate groups in BHIS do not occur. On the other hand, an alkylene carbon attack induces alky-oxygen cleavage to produce mono-alkyl carbonate. Mono-alkyl carbonate is decarboxylated to form CO_2_ (∆ Horsepower (HP) = 112.5 kJ/mol at 170 °C) and it produces ethanol groups in IS [31]. The synthesized BHIS was obtained in a liquid state after Kugelrohr distillation. The chemical composition and structure were analyzed by GC-MS and ^1^H-NMR [32]. The GC-MS chromatogram (Figure 2) confirmed that the mass-to-charge ratio of BHIS was 234 *m/z*. The major chemical components were further confirmed by ^1^H-NMR analysis (Figure 3). In addition, the number of hydroxyl radicals in BHIS was calculated as the average of the three measurements. Table 2 shows the hydroxyl number and molecular weight of BHIS calculated using the method given in Section 2.4. The hydroxyl numbers of the synthesized BHIS were found to be 478.51 mg KOH/g. Using this value and the equation in Section 2.4, the average molecular weight of 234.47 g mol^−^^1^ was obtained. Thus, it was confirmed that BHIS was successfully synthesized.

### 3.2. Preparation of PUs

Bio-based PUs were polymerized through a simple one-shot polymerization without a catalyst, using various ratios of BHIS, PCD, and HDI. M_w_, M_n_, and PDI of the PUs are summarized in Table 3. It was confirmed that M_w_ had an average value of 152,000, whereas the corresponding values for Mn and the polydispersity index were 135,000 and 1.129. Previous studies have created PU using IS. It was confirmed that PU containing a high ratio of IS had a reduced molecular weight [29]. This is because, in polyurethane synthesis, diisocyanate has a different reactivity from IS, which has a secondary alcohol, and PCD, which has a primary alcohol, so the molecular weight is reduced. On the contrary, in the case of PU polymerized with IS substituted with primary alcohol, it can be confirmed that even if the ratio of IS is increased, the molecular weight does not change considerably. Thus, the one-shot polymerization served the purpose, and the PU was successfully polymerized. 

### 3.3. Characterization of PUs by ^1^H-NMR

Figure 4 shows the ^1^H-NMR spectrum of the synthesized PBH PU-2. The peak at 3.21, 1.55, and 1.32 ppm corresponds to the amine proton of the urethane N—H moiety, and the bicyclic methylene proton of isosorbide at 5.47–4.53 ppm. The alkyl chain of isosorbide peaks at 3.7 ppm [29], and the alkylene protons of PCD diol appeared at 4.15, 1.78, and 1.42 ppm [30].

### 3.4. Characterization of PUs by FT-IR Spectroscopy

To demonstrate the formation of PU, FT-IR measurements were performed. The formation of urethane bonds (—NH—COO—) in the FT-IR spectra (Figure 5) was confirmed by the appearance of the characteristic broad N—H peak at around 3332 cm^−1^ in all the samples. The increase in the concentration of IS, which has a smaller molecular weight than that of PCD, shows that both the number of urethane bonds increase, since the urethane bond has an effect on the hydrogen bond, it was confirmed that the peak increased. Furthermore, the carbonyl stretching absorption bands split into two peaks, one at around 1736 cm^−1^ and another at around 1697 cm^−1^. These represent C=O of the carbonate group and C=O of the urethane group, respectively. Asymmetric and symmetric CH_2_ stretching bands were seen in the region around 2938 and 2857 cm^−1^. The bands at 1532 cm^−1^ are assigned to amine II (C—N stretching and N—H bending). The bands at 1236 cm^−1^ are associated with stretching of C—O bonds in Pus. As a result, the characteristic peak N=C=O Peak of HDI was not observed at 2254 cm^−1^, confirming that PU was successfully synthesized [30].

### 3.5. Thermal Properties

It is important to understand the thermal properties of a material, as its thermal properties determine its processing characteristics and applicability. Figure 6 shows the results of the DSC analysis. Two glass-transition temperatures (T_g_) were observed for each DSC curve, indicating the presence of a soft segment and a hard segment. PBH PU-1, PBH PU-2, and PBH PU-3 recorded a glass-transition temperature of −38.33, −37.75, and −32.91 °C, respectively. These are the T_g_ values relevant to the soft segment, which could rotate more freely, leading to the decrease of T_g_. Therefore, the higher the PCD concentration in the soft segment, the lower the T_g_ value. The glass-transition temperature relevant to the hard segment of PBH PU-1, PBH PU-2, and PBH PU-3 was 30.35, 34.14, and 49.91 °C, respectively. The chiral and bicyclic ring structures of the IS and the hard segment, lead to an increase in T_g_ [33].

### 3.6. Mechanical Properties

The tensile properties of the PUs are shown in Figure 7. PBH PU-1, PBH PU-2, and PBH PU-3 were found to be flexible and hard, with tensile strain values ranging from 686.55 to 984.69% and tensile stress values ranging from 33.26 to 63.87 MPa (Figure 7a). The data showed that the tensile stress of PBH PU-3 amounted to 63.87 MPa, the highest tensile stress value (Figure 7b). PBH PU-1 recorded the highest tensile strain of 984.69% (Figure 7c). Furthermore, BHIS/HDI was 890.23 MPa, with the highest Young’s modulus (Figure 7d). These results show that the strength increases as the ratio of BHIS increases due to the chiral and bicyclic ring structure of BHIS rather than the high PCD content of the PU. Furthermore, unlike the strength of conventional bio-based PU, the strength of the PU produced in this study is very high, and the material was confirmed to have excellent physical properties [34,35].

### 3.7. Measurement of Contact Angle

The contact angle was used to study the surface of the material. The sample is measured five times; Figure 8 shows the hydrophilicity and hydrophobicity of the surfaces of PU through a contact angle analysis. Generally, the contact angle with water for a hydrophilic surface ranges from 1 to 30°. For hydrophobic surfaces, the contact angle with the surface is 90° or more. The data show that PCD/HDI = 87.17 ± 1.5°, PBH PU-1 = 72.17 ± 1.2°, PBH PU-2 = 65.95 ± 1.3°, PBH PU-3 = 63.89 ± 1.2°, and BHIS/HDI = 52.39 ± 1.2°. IS is known to be hydrophilic [36,37]. BHIS, with a similar structure, was expected to be hydrophilic. The data showed that the greater the amount of BHIS, the greater its hydrophilicity. The PU produced in this study was able to confirm the hydrophilic properties and was identified as an essential element for application in biomaterials.

### 3.8. Degradation Test Results

In vitro biodegradation experiments of PU were conducted at 37 °C using a phosphate buffer (PBS). All the samples were measured for 56 days. Figure 9 shows the percentage weight loss of the PUs with time. The degradation rates of the PBH PU series started to increase rapidly after 15 days, and a continual slow weight loss of 1.46–3.13% was observed at 56 days. A slow degradation rate was also confirmed in previous PU studies. [30] The BHIS imparts a higher hydrophilicity to the PU segment compared to the PCD unit with hexamethylene moiety, making a hydrolytic attack easier [38].

### 3.9. MTT Assay of Cultured Human Bone Marrow Cells

The cytotoxic efficacy of different formulations of PUs in human bone marrow stromal osteoprogenitor cells (hBMSCs) were assessed by MTT assay at days 1, 3, and 5 (Figure 10). In the results, if the polymer maintains cell growth, it can be considered to have cellular compatibility and biocompatibility. In general, samples with a viability greater than 80–60% compared to controls are considered to be non-cytotoxic [39]. MTT analysis showed that the viability of hBMSCs up to 5 days in the PU series was 112%, 109%, and 115% for PBH PU-1, PBH PU-2, and PBH PU-3, respectively. These values are considered non-toxic. The results of the MTT assay showed that the viability of hBMSCs was well preserved in PBH PU-1, PBH PU-2, and PBH PU-3 up to 5 days, similar to the control. The immunostaining results confirmed that the hBMSCs layer covered the surface of PUs (PBH PU-1, PBH PU-2, and PBH PU-3) similar to the control (Figure 11). Furthermore, on the final day (day 5), all the PUs were found to have grown better than the control cells. These results demonstrate that the produced PUs had good biocompatibility, as it allowed the growth and proliferation of the hBMSCs.

## 4. Conclusions

In this study, we successfully synthesized BHIS containing primary alcohols through a simple synthesis and refined it using IS made from plant sources such as starch, corn, and sugar beet. The bio-based PUs were successfully polymerized through a simple, one-shot polymerization without a catalyst using various ratios of BHIS, PCD, and HDI. The molecular weight was increased, and the polymerization proceeded evenly as compared with PU made of IS. As for the mechanical characteristics, the resulting tensile strain and tensile stress were 984.69% and 63.87 MPa, respectively, indicating that the material is flexible and the strength is good. The produced PU showed excellent cell adhesion in toxicity tests, and the growth of marrow cells was higher than that of the control group on the final (fifth) day, confirming that the material is highly biocompatible. Synthesized BHIS is a compound with increased reactivity compared to IS and can be used to produce bio-based polymers. Thus, the PU polymerized using the approach assessed in this study can be used as a material to reconstruct hard tissues such as bones and can likely function semi-permanently inside a body without immunorejection.

## Figures and Tables

**Figure 1 polymers-15-00418-f001:**
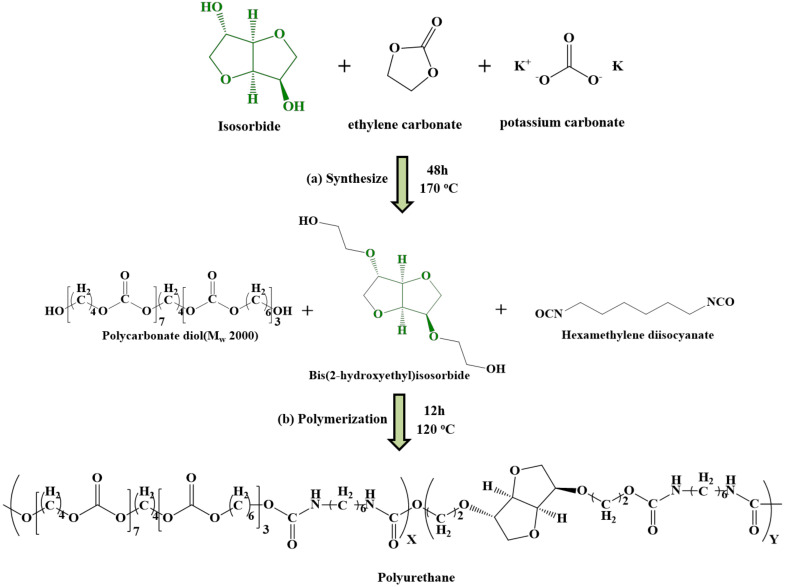
(**a**) Synthetic scheme of BHIS, (**b**) polyurethane structure synthesized with PCD, BHIS, and HDI.

**Figure 2 polymers-15-00418-f002:**
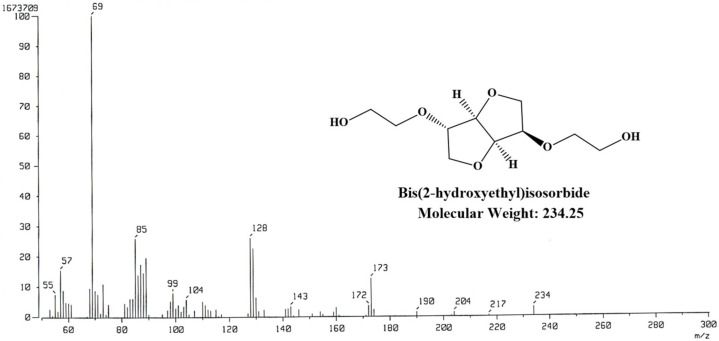
GC-Mass analysis of synthesized BHIS.

**Figure 3 polymers-15-00418-f003:**
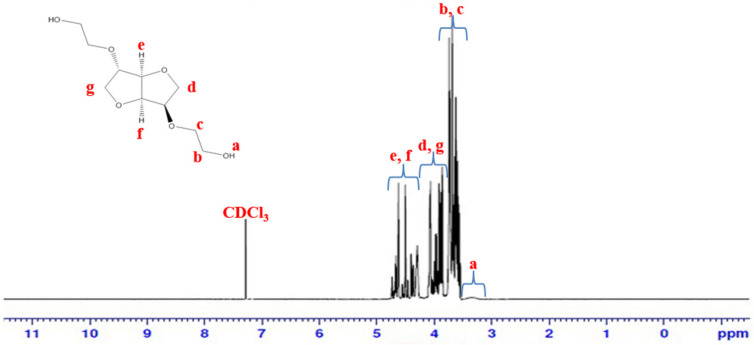
^1^H-NMR analysis of synthesized BHIS.

**Figure 4 polymers-15-00418-f004:**
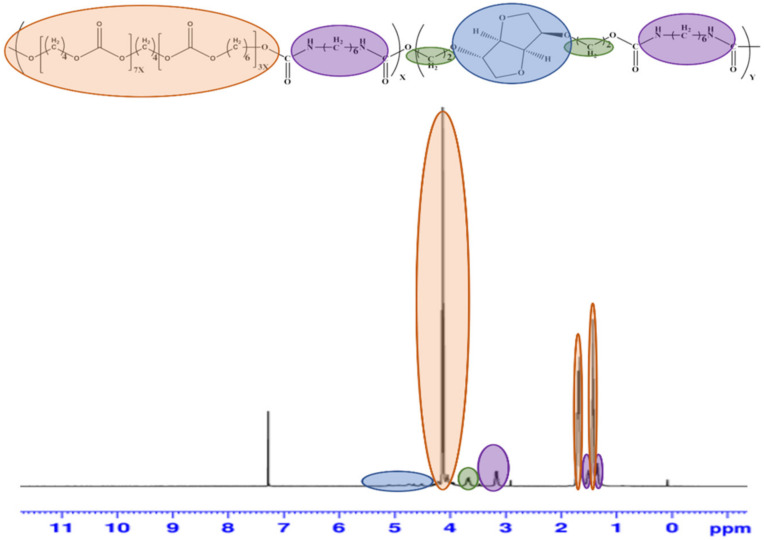
^1^H-NMR spectrum of PU.

**Figure 5 polymers-15-00418-f005:**
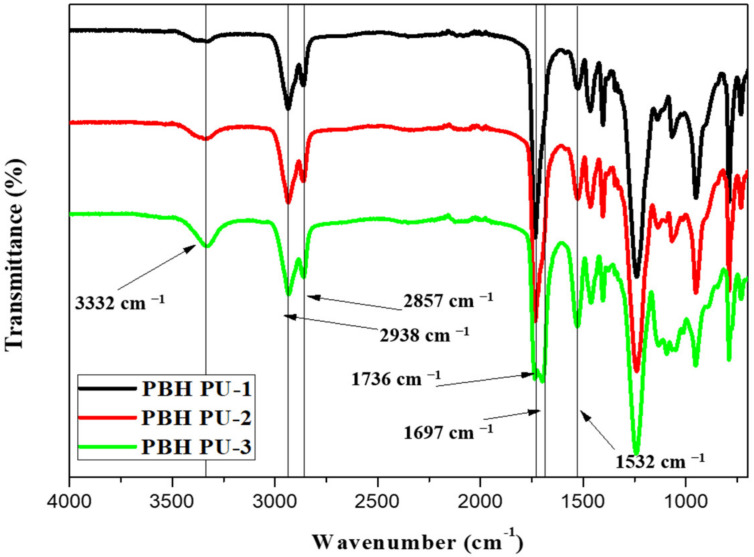
FT-IR spectra data of polyurethanes.

**Figure 6 polymers-15-00418-f006:**
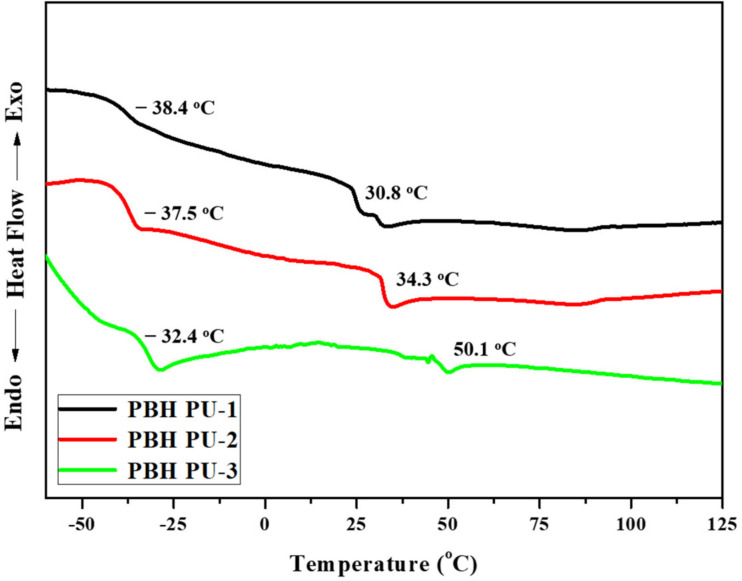
DSC thermograms of polyurethane films.

**Figure 7 polymers-15-00418-f007:**
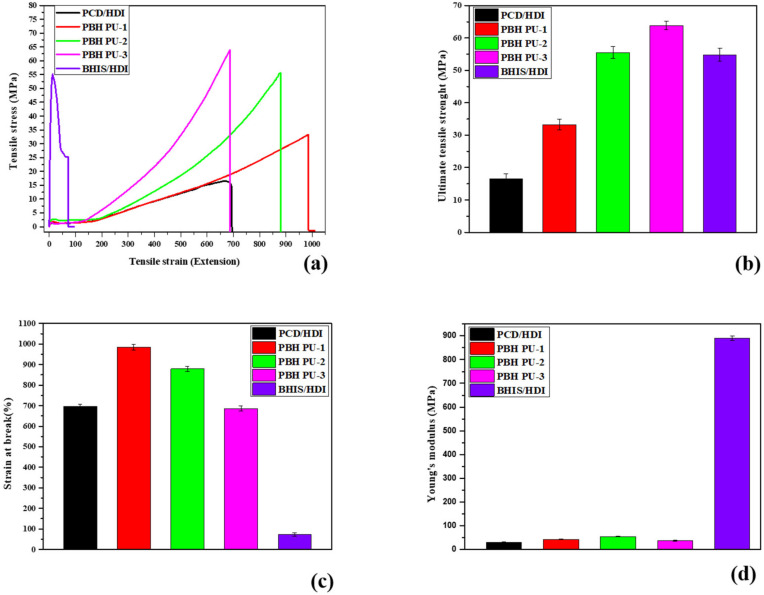
Mechanical property data of polyurethane films. (**a**) Tensile stress, (**b**) ultimate tensile strength, (**c**) strain at break, and (**d**) Young’s modulus.

**Figure 8 polymers-15-00418-f008:**
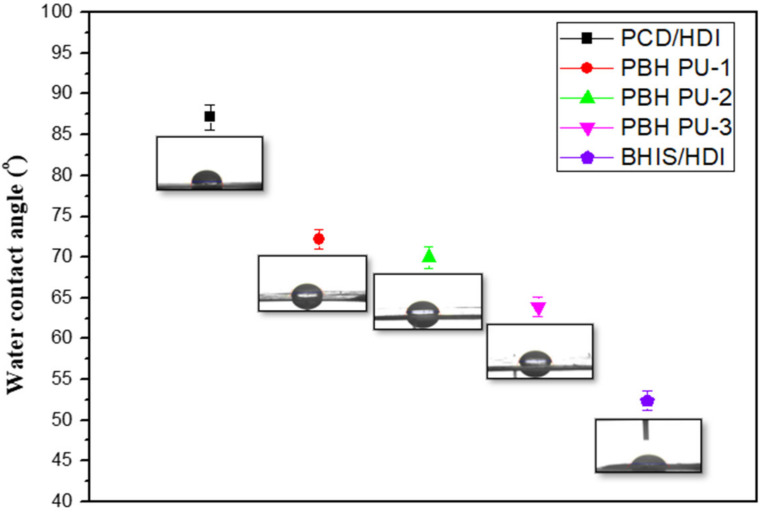
Contact angle images of polyurethanes.

**Figure 9 polymers-15-00418-f009:**
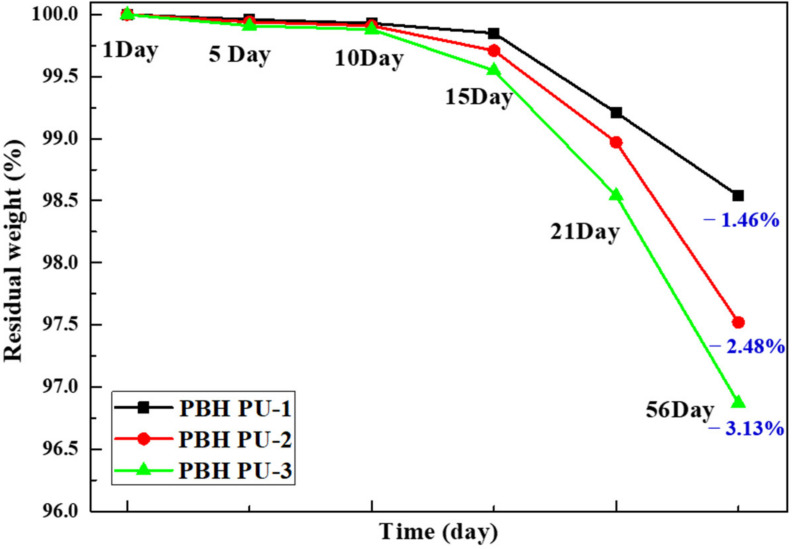
Degradation test data of polyurethane films.

**Figure 10 polymers-15-00418-f010:**
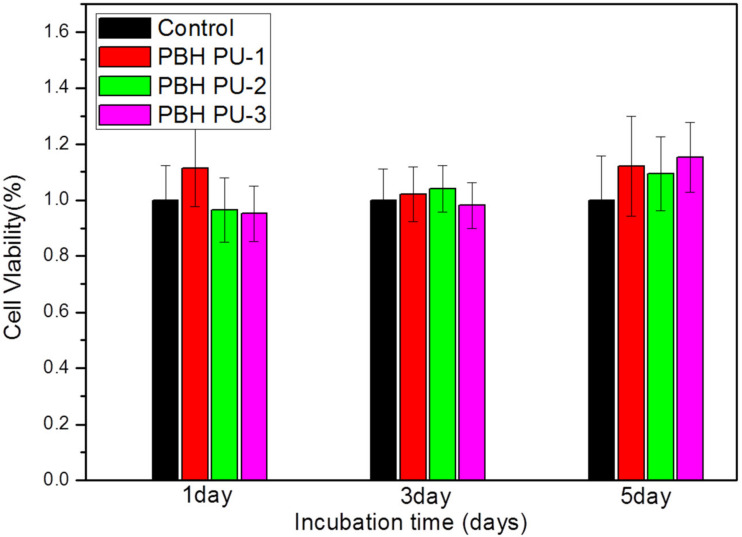
MTT assay of hBMSCs cultured on polyurethanes.

**Figure 11 polymers-15-00418-f011:**
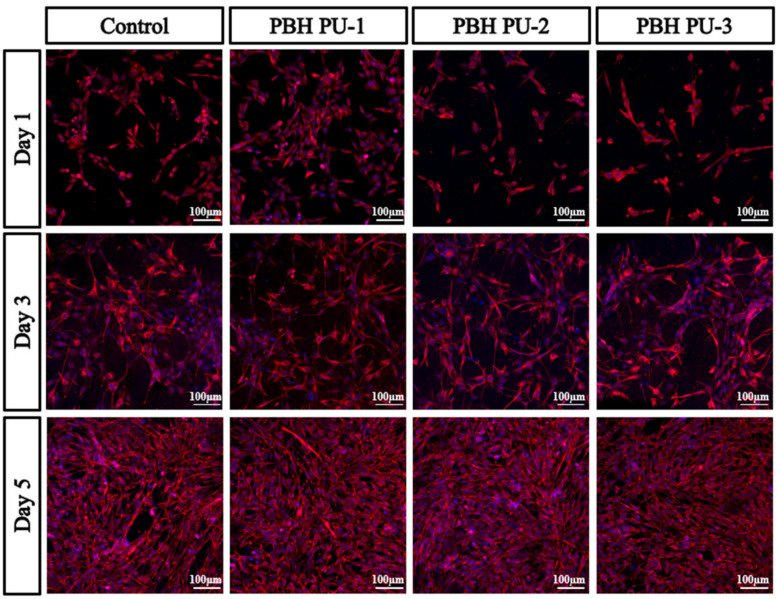
Confocal image of immunocytochemistry staining with F-actin (red) and nuclei (blue) of the adherent cells on the PCD films at 1, 3, and 5 days.

**Table 1 polymers-15-00418-t001:** Polyurethane series with different PCD and contents.

Polyurethanes	HDI	PCD	BHIS	Yield (%)
mole ratio/weight (g)				
PCD/HDI	1/(10)	1/(118.913)	0/(0)	95.9
PBH PU-1	1/(10)	0.8/(95.131)	0.2/(2.786)	96.8
PBH PU-2	1/(10)	0.5/(59.457)	0.5/(6.964)	97.5
PBH PU-3	1/(10)	0.2/(23.783)	0.8/(11.142)	97.8
BHIS/HDI	1/(10)	0/(0)	1/(13.928)	98.1

**Table 2 polymers-15-00418-t002:** Hydroxyl numbers and molecular weight of BHIS.

Polymer	Hydroxyl Numbers (mg KOH/g)	Targeted M_n_ (g/mol)
BHIS	478.51	234.47

**Table 3 polymers-15-00418-t003:** GPC data of polyurethane series.

Polyurethanes	M_w_ ^1^	M_n_ ^2^	PDI ^3^
PBH PU-1	152,941	136,874	1.117
PBH PU-2	150,318	130,845	1.149
PBH PU-3	154,181	137,501	1.121

^1^ The weight average molecular weights. ^2^ The number average molecular weights. ^3^ Polydispersity index. M_w_/M_n._

## Data Availability

Not applicable.

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
