# Peer review of "Synthesizing Polyurethane Using Isosorbide in Primary Alcohol Form, and Its Biocompatibility Properties"

_polymers, 2023, doi:10.3390/polym15020418_

Round 1

Reviewer 1 Report

In this paper the authors have synthesized a sorbitol based biodegradable PU. The authors have synthesized IS substituted with primary alcohol (BHIS) and synthesized PU with BHIS, PCD and HDI. The authors have  also shown the excellent mechanical properties of the synthesized PU. The results are well represented and the paper can be accepted after minor revisions

1. The last 5 lines of abstract requires revision since it has grammatical errors 

2. What was the rationale behind using the mole ratios used to synthesize the 3 different PUs

3. Table 1 must be edited for better clarity. It is confusing whether it is the weight in grams which is in brackets or the mmoles used.

Reviewer 2 Report

The manuscript deals with the synthesis of degradable polyurethanes based on a biobased component, namely OH-modified isosorbide.

Modification of isosorbides by ethylene carbonate leads to primary OH functions, which can be well used as hard segment components in polyurethanes. The successful synthesis was characterized by NMR, GC-MS and OH end group determination.

OH-modified isosorbide was used in various concentrations with a polycarbonate diol and hexamethylene diisocyanate to synthesize polyurethanes. The synthesized polyurethanes were processed into films. Material characterization was performed by NMR, IR, DSC and contact angle measurements. Biocompatibility was investigated by cell seeding and immunostaining.

In summary, this is an interesting and relevant work. The investigations performed are very comprehensive and timely.

However, before publication, there are still some points that need to be considered from my point of view:

1) Headline: "Synthesizing bio-based polyurethane...". The title is not quite appropriate, since only one component is bio-based and this component was also modified with ethylene carbonate. In this respect, the title is somewhat misleading.

2) Line 21: Typo (born bone marrow cells)

3) Materials and Methods: Abbreviations have already been explained in the introduction, but it is appropriate to write out the abbreviations again for the material part (PCD, HDI, EC etc).

4) Line 82: Please change to: 2.2. synthesis of bis(2-hydroxyethyl)isosorbide (BHIS)

5) Line 86: here it says: "After cooling, the column was purified ..." Where did the column suddenly come from. What was it used for? What dimensions, what stationary phase? It is not clear.

6) Line 112: Here it says: "After purification,...", What exactly is meant here? What was done as an additional purification step?

7) Figure 1: The figure is very meaningful but the letters in the scheme are unfortunately a bit too small. Furthermore, no X should appear in the PCD, since the Mw should be 2000, X can be calculated.

8) Line 146: What are the dimensions of the samples in dumbbell shape? Are the data comparable with data according to ISO standard? A known reference polyurethane would be nice for comparison.

9) Line 149: Degradation test: Please provide underlying literature.

10) Line 166: DPBS is mentioned for the first time, please write out.

11) Line 179: FITC phalloidin, please write out.

12) Line 189: What do the brackets (1) and (2) mean?

13) Lines 191-197: The wording of the mechanism is not understandable. A formula scheme would be helpful (such as in Ind. Eng. Chem. Res. 2003, 42, 663-674).

14) Figure 3 page 7: NMR evaluation: the data of the integrations would be helpful to estimate the purity. The cited literature [29] is from 2020 not 2019 (ACS Biomaterials Science & Engineering 2020 6 (5), 2578-2587).

15) Line 221: The sentence is not understandable. Probably: ... relatively slow in comparison to PCD.

16) Figure 4, page 8: Subtitle: Which of the three PUs has been studied here by NMR? The drawing is much too small. Maybe the section from 0-7 ppm should be shown. As the figure is now, it does not provide any useful information.

17) Line 242: The beginning of the sentence is missing.

18) Line 246: Typo “Since” small letter

19) Figure 6: In each case the first heating curve is shown, which reflects the thermal history of the sample. The second heating curve therefore should also always be given.

20) Lines 295-298: The statement of the sentence should be formulated more clearly.

21) Figure 8, page 10: What is the number of measurements? I could not find.
